# An Analysis of Agricultural Production Efficiency of Yangtze River Economic Belt Based on a Three-Stage DEA Malmquist Model

**DOI:** 10.3390/ijerph19020958

**Published:** 2022-01-15

**Authors:** Zhiwei Pan, Decai Tang, Haojia Kong, Junxia He

**Affiliations:** 1School of Management Science and Engineering, Nanjing University of Information Science & Technology, Nanjing 210044, China; 20201224017@nuist.edu.cn (Z.P.); 20211900022@nuist.edu.cn (J.H.); 2China Institute of Manufacturing Development, Nanjing University of Information Science & Technology, Nanjing 210044, China; 3School of Economics and Management, Nanjing University of Science & Technology, Nanjing 210000, China

**Keywords:** DEA model, Malmquist index, Yangtze River Economic Belt, agricultural production efficiency

## Abstract

The Yangtze River Economic Belt (YREB) is a major national strategic development area in China, and the development of the YREB will greatly promote the development of the entirety China, so research on its agricultural production efficiency is also of great significance. This paper is committed to studying the agricultural production efficiency of 11 provinces in the YREB and adopts a combination of the Data Envelopment Analysis (DEA) model and the Malmquist index to make a dynamic and static analysis on the YREB’s agricultural production efficiency from 2010 to 2019. Then, a three-stage DEA Malmquist model that eliminates the factors of random interference and management inefficiency is compared to a model without elimination. The results show that the adjusted technological efficiency changes, technological progress, and total factor productivity increased by −0.1%, 0.24%, and 0.22%, respectively. When comparing these values to the pre-adjustment values, the results indicate that the effect of environmental variables cannot be ignored when studying the agricultural production efficiency of the YREB. At the same time, the differences in the agricultural production efficiency in the YREB are reasonably explained, and feasible suggestions are put forward.

## 1. Introduction

China is an advanced agricultural country, as it forms an important foundation for economic development, social progress, and industrial structure adjustment [1]. Therefore, the state has also issued a series of policies for the processes of agricultural development to promote the sustainable and healthy development of agriculture. The Yangtze River Economic Belt (YREB) includes 11 provinces and cities: Shanghai, Jiangsu, Zhejiang, Anhui, Jiangxi, Hubei, Hunan, Chongqing, Sichuan, Guizhou, and Yunnan (Figure 1). Amongst these provinces/cities, there are great differences in economic development level, industrial structure, and agricultural production efficiency [2,3]. There are numerous studies on the YREB. Wang et al. studied the emission reduction efficiency of PM2.5 in the YREB, which is of great significance to the control of atmospheric environmental pollution [4]. Zhang et al. studied the relationship between the economic development and the pressure of the water environment in the YREB [5]. Liu et al. analyzed the spatial distribution characteristics and driving forces of total phosphorus emissions in the YREB, which is of great significance to the treatment of phosphorus pollution in the YREB [6]. In addition, agricultural issues in the YREB have also become an important research field. In 2018, the state issued the *Strategic Plan for Rural Revitalization (2018–2022)* proposal, which was meant to build a modern agricultural industrial system, realize the integrated development of rural primary, secondary and tertiary industries, promote rural strategic transformation, and enhance the innovation and competitiveness of China’s agricultural industries. As an important development strategy of the country, the YREB is also of great significance to the nation’s agricultural economic development and agricultural production efficiency (APE). Based on the dynamic and static analysis of agricultural production efficiency in the 11 provinces included the YREB, this paper aims to study the regional differences in agricultural development levels in the YREB territories and to put forward feasible suggestions for the current agricultural development situation in this region and China. The innovation of the presented paper is that a three-stage DEA model is used to study the agricultural efficiency of the YREB, which eliminates environmental impact and random impact, meaning that the results are more objective. In addition, spatio-temporal analysis of agricultural efficiency can dynamically reflect the changing characteristics of agricultural efficiency in the YREB.

This paper is structured as follows: Section 2 contains the literature review, Section 3 contains the research methodology and data, Section 4 contain experimental procedure, and Section 5 and Section 6 contain the discussion and conclusion, respectively.

## 2. Literature Review

APE evaluation is very important to a country’s agricultural development level, and there are a variety of ways to measure it. For example, Shah et al. used energy accounting to analyze the sustainability of Pakistan’s agricultural production system, helping decision makers understand the important role of ecosystems in agricultural production systems [7]. Guo et al. used labor, land, and capital as input variables and agricultural economic benefits as output variables. There, they used the Stochastic Frontier Approach (SFA) model, spatial correlation analysis method, and Tobit model to effectively evaluate the spatial layout of agricultural production and APE optimization [8]. Li et al. used the data envelopment analysis (DEA) model to estimate the technical efficiency and energy-saving potential of China’s 30 provincial agricultural departments from 1997–2014 and concluded that there are differences in APE in the eastern and western regions of China [9]. Agricultural efficiency is also inseparable from the sustainable development of agriculture. As a case in point, Laurett et al. used an exploratory factor analysis (EFA) model to explore the important factors affecting the sustainable development of agriculture [10]. Valizadeh and Hayati used factor analysis to measure the indicators of sustainable agricultural development [11]. Shen et al. decomposed the overall inefficiency into three components of technology and the mixing and scale effect and conducted empirical research on the economic and environmental performance and APE of 31 provinces in China from 1997–2014 [12]. Wang et al. used multilateral total factor growth rate estimates to measure the differences in the production efficiency of the agricultural sector in various regions of China [13]. Ma et al. used spatial autocorrelation and econometric models in an analysis of China’s economic value from 1990–2017 and found that China’s APE is generally low [14].

In the research on agricultural efficiency and factors affecting agricultural production, the data envelopment analysis (DEA) model is one of the most widely used methods. Based on the panel data from 2002 to 2015, Yu and Zhang used the ultra-efficient DEA model and Malmquist index to measure and analyze the APE of Shandong Province, China [15]. Li et al. used panel data from 30 administrative regions in China as the research objects, selected a series of indicators as input and output variables, and used the DEA model to analyze the agricultural total factor production efficiency of various provinces from 1997–2014 and concluded that China’s agricultural total factor productivity showed a slow growth trend in fluctuation [16]. In recent years, in order to evaluate agricultural efficiency more effectively, some scholars have used and developed DEA models. Mosbah et al. converted the DEA model into two twin input decomposition and output decomposition (i.e., ID-DEA and OD-DEA) models and effectively evaluated agricultural efficiency [17]. Toma et al. estimated the APE of EU countries from 1993 to 2013 based on the non-parametric Bootstrap DEA model and concluded that from the perspective of resource conservation that the oldest EU countries have higher APE [18]. Based on the global DEA model and the weighted Russell distance function model, Liu and Feng used 30 provinces in China as a sample to analyze green total factor productivity in the period from 2005 to 2016 and to propose countermeasures for the development of green agriculture in China [19].

Another mainstream method for studying APE is the SFA. To demonstrate this, Deng and Gibson used the Land Production Estimation System (ESLP) to estimate the land productivity of Shandong Province in the period from 1990–2010 and analyzed the ecological efficiency of sustainable agricultural production based on SFA [20]. Yigezu et al. used the SFA model to measure the efficiency of agricultural irrigation water in Syria [21]. Ilaria et al. evaluated agricultural efficiency using the heteroscedasticity stochastic frontier model with Italian farm characteristics as input variables [22]. Villano and Fleming applied the SFA model to a technical efficiency analysis of rice production and evaluated its production risks [23].

It can be seen that the DEA method can be used to evaluate the production or business performance of decision-making units with multiple inputs and multiple outputs. The DEA method does not need to specify the form of the production function of the input and output, so it can evaluate the efficiency of decision-making units (DMU) with more complex production relations [24,25]. In summary, this paper uses a combination of a three-stage DEA model and the Malmquist index to study agricultural efficiency. According to the experience of some scholars [26,27,28,29], this paper takes the sown area of crops, the number of fertilizers, the agricultural practitioners, the total power of agricultural machinery and the effective irrigation area as input variables and the total agricultural output value and the output of main crops as output variables. At the same time, considering factors such as random disturbance and management inefficiency, this paper incorporates the affected crop areas, government subsidies, and gross regional product as environmental variables into the model system, and scientifically analyzes and evaluates the APE of the YREB (Figure 2) [30,31].

## 3. Data and Methodology

### 3.1. DEA Model

DEA is very popular model for dealing with multi-input and especially non-single output [32]. This method can evaluate the relative effectiveness of decision-making units, pays attention to optimizing each decision-making unit, and provides directional guidance for the adjustment of relevant indicators. The more mature CCR model and the BCC model are more frequently used in DEA. BCC assumes that the variable returns to scale, and the model form is as follows:(1)$$s.t.\left\{ \begin{array}{l} \min\theta \\ \sum_{j=i}^{n}\lambda_{j}x_{ij}+s^{+}=\theta x_{i0},i=1,2,\ldots ,m\\ \sum_{j=i}^{n}\lambda_{j}\gamma_{ij}+s^{-}=\theta \gamma_{\gamma 0},\gamma =1,2,\ldots ,m\\ \lambda_{j}\geq 0,j=1,2,\ldots ,n\\ \sum_{j=1}^{n}\lambda_{j}=1\\ s^{+}\geq 0,s^{-}\leq 0 \end{array}\right.$$
where θ is the efficiency value of the DMU decision-making unit, $\lambda_{j}$ represents the combination ratio of the *j*-th decision-making unit DMU in a certain effective DMU combination, $x_{ij}$ is the total input of the *j*-th DMU to the *i*-th input, and $\gamma_{ij}$ is the *j*-th DMU. For the total output of the *i*-th category output, $S^{+}$ and $S^{-}$ represent the slack variable. If θ  = 1, $S^{+}=S^{-}$ = 0, then the DEA of the decision-making unit is valid, and it is on the leading edge. If θ  = 1, $S^{+}$ ≠ 0, or $S^{-}$ ≠ 0, the DEA of the decision-making unit is weakly effective. If θ  < 1, then the DEA decision unit is not valid. The calculated efficiency value is comprehensive technical efficiency (TE), which can be further decomposed into scale efficiency (SE) and pure technical efficiency (PTE).

### 3.2. Malmquist Index

Static DEA analysis can only analyze the efficiency value horizontally and cannot show a dynamic trend. In order to reflect the agricultural development of the YREB more comprehensively and intuitively, based on the static analysis, this paper introduces the Malmquist index method, which deeply analyzes the agricultural development through the changes in the cross-period efficiency value and then puts forward targeted countermeasures to promote the balanced development of agriculture in all of the YREB provinces. The Malmquist index solution process is as follows:(2)$$\begin{array}{l} M(x_{t},y_{t},x_{t}{}_{+1},y_{t+1})={\left[\frac{D_{c}^{t}(x_{t}{}_{+1},y_{t+1})}{D_{c}^{t}(x_{t},y_{t})}*\frac{D_{c}^{t+1}(x_{t}{}_{+1},y_{t+1})}{D_{c}^{t+1}(x_{t},y_{t})}\right]}^{\frac{1}{2}}\\ =\frac{D_{c}^{t+1}(x_{t}{}_{+1},y_{t+1})}{D_{c}^{t}(x_{t},y_{t})}*{\left[\frac{D_{c}^{t}(x_{t}{}_{+1},y_{t+1})}{D_{c}^{t+1}(x_{t}{}_{+1},y_{t+1})}*\frac{D_{c}^{t}(x_{t},y_{t})}{D_{c}^{t+1}(x_{t},y_{t})}\right]}^{\frac{1}{2}}\\ =effch*techch \end{array}$$
where $x_{t}$ and $x_{t+1}$ are the input variables, $y_{t}$ and $y_{t+1}$ are the output variables, and $D_{c}$ is a distance function. EFFCH represents changes in the technological efficiency, TECHCH represents technological change, and EFFCH can be decomposed into the product of pure technological efficiency change (PECH) and scale efficiency change (SECH).

### 3.3. Stochastic Frontier Approach

The Stochastic Frontier Approach (SFA) can eliminate the impact of environmental variables and statistical noise on the effectiveness of a decision-making unit. Therefore, in the second stage, the stochastic frontier model is mainly used to exclude environmental variables and statistical noise, leaving only relaxation variables created by management inefficiency.

Step 1:(3)$$S_{ni}=f(Z_{i},\beta_{n})+\gamma_{ni}+\mu_{ni}$$

Step 2:(4)$$X_{ni}^{A}=X_{ni}+[\max(f(Z_{i},{\hat{\beta }}_{n}^{}))-f(Z_{i},{\hat{\beta }}_{n}^{})]+[\max({\hat{\gamma }}_{ni})-{\hat{\gamma }}_{ni}]$$
where, $s_{ni}$ represents the slack value of the decision-making unit, $Z_{i}$ represents the environmental variables, $\beta_{n}$ is the environmental variable coefficient, $\gamma_{ni}$ represents random interference items, and $\mu_{ni}$ represents management inefficiency. $X_{ni}^{A}$ represents the input variable after adjustment, $X_{ni}$ represents the input variable before adjustment, $max\left(f\left(Z_{i},{\hat{\beta }}_{n}\right)\right)-f\left(Z_{i},{\hat{\beta }}_{n}\right)$ represents the adjustment of all of the decision-making units to the same external environment, and $max\left({\hat{\nu }}_{ni}\right)-{\hat{\nu }}_{ni}$ represents the adjustment of the random errors of all of the decision-making units to the same situation.

### 3.4. Data

In this work, data are derived from the following sources: The China Statistical, City Statistical and Provincial Statistical Yearbooks from 2011–2020. The data used include the total sown area of crops, agricultural employees, agricultural fertilizer usage, total machinery power, effective irrigation area, disaster areas, agricultural subsidies, and regional GDP. Missing data were calculated via the interpolation of the adjacent year.

## 4. Agricultural Efficiencies of Each Province in the YREB

### 4.1. Static Analysis and Temporal and Spatial Evolution of Agriculture in the YREB

According to the provincial agricultural data from the YREB, this paper selects the sowing area of crops, the amount of chemical fertilizer, agricultural employees, the total power of agricultural machinery, and effective irrigation area as input variables, and the total agricultural output value and the output of main crops as the output variables. The APEs in 2010 and 2019 were analyzed using DEAP 2.1 software, and the following results are obtained: Table 1, Table 2 and Table 3.

It can be seen that the technical efficiency, pure technical efficiency, and scale efficiency of the provinces in the YREB are all on the rise, with the average values increasing by 3.7%, 2%, and 1.7%, respectively. In 2010, there were three provinces (Jiangxi, Yunnan, and Guizhou) with increasing returns to scale in the YREB, three provinces (Hubei, Zhejiang, and Hunan) with diminishing returns to scale, and five provinces (Jiangsu, Anhui, Sichuan, Chongqing, and Shanghai) with constant returns to scale. By 2019, only Hunan had increasing returns to scale, with an additional three provinces (Hubei, Zhejiang, and Yunnan) with diminishing returns to scale, and seven provinces (Jiangsu, Anhui, Sichuan, Chongqing, Shanghai, Jiangxi, and Guizhou) with constant returns to scale. In order to express the temporal and spatial evolution of agricultural efficiency more intuitively in the YREB, this paper used ArcGIS 10.2 to plot the changes in the technical efficiency, pure technical efficiency, and scale efficiency in each province in 2010 and 2019, as shown in Figure 3, Figure 4 and Figure 5.

### 4.2. The First Stage DEA-Malmquist

According to the principles and calculations of Equations (1) and (2), which were made using the DEAP 2.1 software, the initial APE evaluation results for each province in the YREB (Table 2) and the total efficiency changes in the YREB from 2010–2019 were obtained (Table 3).

(I) From the perspective of technological progress, with the exception of the fact that the technological progress in the YREB from 2011 to 2012 was less than 1, the technological progress of other years was greater than 1, indicating that the technological progress of the YREB was generally at a high level, which promoted total factor productivity progress. Looking at each province individually, only Jiangsu, Hubei, and Jiangxi have technological progress less than 1 from 2010–2019. Other provinces have technological progress that is greater than 1, and the overall average value is 1.012, which represents a relatively high level. (II) On the whole, from 2010–2019, the APE index of the YREB has alternately increased and decreased, but the efficiency index has become more and more stable, and there is an overall increasing trend. With the exception of the efficiency values of Jiangsu and Hubei provinces, which are less than 1, the efficiency values of the other provinces are all greater than 1. It can be seen that the main driving factor of APE in the YREB is technological progress. In order to see the changes in the total factor productivity in the YREB more intuitively, this article uses a histogram to show it more clearly (Figure 6):

### 4.3. SFA Regression in the Second Stage

Table 4 shows the impact of the environmental variables on agricultural production-related inputs. As shown in the table, environmental variables have an impact on the five input factors, and most of them are positive. The γ values of the five input slack variables are 0.64, 0.61, 0.05, 0.71, and 0.67, respectively. Four of them are greater than 0.6 and are at a relatively high level, indicating that the environmental variables have a significant impact on their input indicators, and the LR likelihood tests all passed the 1% significance level. Therefore, it is necessary to study the impact of environmental variables on input and to eliminate them to make the research results more objective and accurate. When the impact coefficient of the environmental variables on the input variables is positive, such as the impact coefficient of disaster areas on crop sowing area being (0.027), which is positive, it shows that increasing the number of environmental variables will reduce the difference variable; that is, the impact of disaster area on APE will be negative. When the influence coefficient of the environmental variables on the input variables is negative, such as when the influence coefficient of regional GDP on total mechanical power is (−0.011), it indicates that increasing the number of environmental variables will increase the difference variable; that is, an increase in the regional GDP will have a positive impact on APE.

### 4.4. The Third Stage of DEA-Malmquist

After SFA regression in the second stage, the input variables are adjusted by Equation (4), and then DEA analysis is carried out according to the same method used in the first stage to obtain the adjusted APE of the provinces in the YREB in period from 2010–2019. The results are presented in Table 5 and Table 6. 

Similarly, the adjusted APE of the YREB will be evaluated in three aspects. (I) From the perspective of changes in technical efficiency, the overall technical efficiency changes in the YREB from 2010 to 2019 show a downward trend, and the changes in technical efficiency in various provinces fluctuate around 1, which is also because the pure technical efficiency and scale efficiency are basically unchanged. (II) From the perspective of technological progress, the technological progress of the YREB is almost greater than 1, which is also the main driving force to promote changes in total factor productivity. However, from 2010 to 2019, the technological progress of the YREB changed from an increase to a decrease, but it remained at generally high levels. (III) As for the total factor productivity of the YREB, the change in the total factor productivity also shows an alternating change of a rise and fall from 2010 to 2019, which was mainly affected by technological progress. It can be seen that the technological progress of the YREB directly affects changes in the total factor productivity and plays a leading role [33,34,35].

## 5. Results and Discussion

Compared to the first stage, the changes in the technical efficiency, technological progress, and total factor productivity after adjustment increased by −0.1%, 0.24%, and 0.22%, respectively, indicating that the impact of environmental variables cannot be ignored and that they will have a great impact on the APE. As discussed by Fei [36] and Li [37], agricultural production is affected by various environmental factors. Among them, the change in technical efficiency has a negative effect on the efficiency of agricultural production. Specifically, the technical efficiency of Hunan, Hubei, and Shanghai is less than 1, representing a low level, resulting in low total factor productivity. Therefore, these cities should focus on improving technical efficiency. The total factor productivity of Zhejiang, Yunnan, and Guizhou is higher than the average value of the YREB, indicating that there is no large input redundancy phenomena in these three provinces, and the APE is high.

DEA analysis was conducted for the adjusted input–output variables for 2019. Taking technical efficiency as the abscissa and scale efficiency as the ordinate, the decomposition diagram of the comprehensive efficiency of the 11 provinces and cities in the YREB was determined and was divided into four categories, as shown in Figure 7 and Table 7. The first category is a double low type of technical efficiency and scale efficiency, which can be seen in Shanghai, Chongqing, and Hunan. These three cities need to continue to improve their technical capabilities and technical levels to improve their local APE. The second and third categories include Hubei Province, which has a low technical level, and Jiangxi Province, which as low scale efficiency. These two cities have some deficiencies in terms of technical efficiency and scale efficiency, respectively, so they can learn from each other. The fourth category is a double high type of technical efficiency and scale efficiency, and these are represented by Jiangsu, Anhui, Sichuan, Zhejiang, Yunnan, and Guizhou. If these cities want to further improve their agricultural efficiency, then they must develop new technologies and innovative agricultural means, such as the introduction of new agricultural technologies, the promotion of agricultural mechanized production, the reduction of as many agricultural and human costs as possible, and the optimization of the layout of the agricultural industry.

Throughout the YREB, the APE of provinces and cities is generally at a high level because the YREB is a major strategic area for national development. Promoting the development of agricultural production in the YREB; forming a pattern of complementary advantages, cooperation, and interaction between upstream, middle, and downstream areas; and narrowing the development gap between the east, central, and western regions is conducive to a new path towards innovative and green agricultural development. Therefore, according to the current agricultural development situation in the YREB, this paper puts forward the following suggestions: First, the government should implement the strategy of differentiated development to promote the balanced development of regional agriculture, especially in areas with low agricultural efficiency, such as in Jiangsu, Hubei, Shanghai, and so on. These areas should improve production conditions as guidance, strengthening deep cultivation and fine farming ability, developing fine agriculture, tapping into the characteristic agricultural products with great potential and high efficiency, and improving product quality and influence [38,39]. At the same time, these areas need to introduce excellent varieties, build high-standard farmland, cultivate high-quality main production areas, and realize the multi-layer superposition of benefits. In addition, the government should strive to improve the technical level of these areas, promote agricultural innovative development, and narrow the gap in agricultural development among regions. Second, the government needs to improve the level of regional financial support for agriculture. Local financial expenditure on agriculture, forestry, and water has a positive effect on agricultural efficiency. Agricultural subsidies in Chongqing, Shanghai, and Jiangxi are low, so they also have a certain impact on the improvement of agricultural efficiency. Therefore, on the one hand, the agricultural sector should strengthen its financial support for agriculture in order to improve the government’s attention to agricultural development. On the one hand, the state council should also standardize the management and process acquiring agricultural support for agriculture and should strengthen the construction of corresponding laws and regulations to ensure that the support funds are scientific, reasonable, and smooth. Third, the provinces located in the YREB may be affected by natural disasters, resulting in the APE not reaching the expected level, especially in provinces with high agricultural sowing areas, such as Yunnan, Hunan, and Hubei. The government should pay attention to the affected area and scope of crops, improve agricultural policies, and introduce financial and insurance services to the agricultural sector, minimize losses in the process of agricultural development, and improve APE. Fourth, it is necessary to deepen the reform of the government system, strengthen government supervision, and rationally allocate agricultural resources. Even though the industry sector plays a leading role in agricultural product planting, processing, tourism, and other industries, it should also focus on the joint development of rural primary, secondary, and tertiary industries, the innovation of new models of agricultural development, the creation of regional advantages, and build core competitiveness in local agricultural development [40,41].

## 6. Conclusions

As it is an important national development strategy, the research on agricultural efficiency is of great economic significance. This paper makes a dynamic and static analysis on the agricultural efficiency of the YREB from 2010–2019 through a three-stage DEA Malmquist model. First, this paper makes a static analysis on the agricultural efficiency of provinces in the YREB and clearly shows the differences in the APE through the use of a spatiotemporal evolution diagram. Then, it provides a dynamic analysis on the APE of the YREB from 2010–2019 and studies the differences in the agricultural efficiency in the YREB by using the combination of the DEA and Malmquist index without excluding the influencing factors of environmental factors and random disturbance, with the APE of the YREB showing an increasing trend in terms of fluctuation=; however, the APE of Jiangsu and Hubei is low due to environmental factors. Then, SFA was used to eliminate the impact of environmental variables and statistical noise on the effectiveness of the decision-making unit in the second stage. The environmental variables include affected area, regional GDP, and agricultural subsidies. The results of the stochastic frontier show that the γ value and LR unilateral detection are significant. Therefore, it is necessary to eliminate environmental factors and statistical noise factors to further analyze the APE of the YREB. Finally, the third stage DEA Malmquist model is used to study the APE of the YREB. It was determined that the agricultural development level of the entire YREB is better. Compared to the results of the first stage, environmental variables affected the development of agricultural production to a certain extent. After excluding the influencing factors of the environmental variables, this paper was divided into four categories according to the differences in the agricultural efficiency values among the provinces and cities in the YREB and then put forward problems and guiding suggestions for agricultural development according to the different categories. In short, in-depth research on regional agricultural efficiency is beneficial for the government to make effective policy, which can promote the rapid agricultural development.

Based on these results and discussions, changes and drivers of agricultural efficiency can be further analyzed:

(a) The overall agricultural production efficiency of Shanghai, Chongqing, and Hunan is relatively low. The reason for this is that Shanghai and Chongqing are municipalities that are directly under the central government of the YREB, and their geographical area is relatively small, resulting in a decrease in the sown area of crops and the number of farmers. Hunan is located in the middle reaches of the YREB, and natural disasters are relatively serious, resulting in low agricultural efficiency, which is also demonstrated in the observation data.

(b) Although Jiangxi’s agricultural scale efficiency is relatively good, its overall efficiency is relatively low, and the total agricultural output value is not high. Because Jiangxi has more agricultural planting areas but less local agricultural subsidies, farmers lack high levels of enthusiasm and motivation. The total agricultural factor productivity is low. Therefore, government subsidies are also crucial for the improvement of agricultural production efficiency.

(c) The reason for the low agricultural efficiency in Hubei is technical efficiency. The agricultural irrigation technology and machine power in this province are relatively low compared to other provinces and cities. Therefore, in order to improve agricultural production efficiency in Hubei, it is necessary to improve the local technical level.

(d) The agricultural production efficiency of Jiangsu, Anhui, Zhejiang, Sichuan, Yunnan, and Guizhou were able to reach optimal levels. The internal factors permitting this are the superior geographical location, high agricultural scale efficiency, and advanced agricultural technology that these provinces. The external factors are that these provinces have a low frequency of natural disasters and high government subsidies. Therefore, the input–output ratio of agriculture was able to reach a reasonable state. If these areas want to further improve their agricultural production efficiency, they need to implement innovative agricultural development strategies.

APE represents the degree of agricultural development. According to the above conclusions some suggestions to realize the development of agricultural production are discussed below:

**Increase infrastructure construction and consolidate the foundation of agricultural development.** First, the agricultural production sector should calculate the sowing area, production conditions, and resource distribution of local agriculture and should adjust the agricultural scale on the premise of ensuring reasonable output. Second, provinces and cities in the YREB should strengthen the construction of farmland and water conservancy facilities, carry out production technologies such as intelligent irrigation and energy-saving drainage, and improve production conditions and further consolidate the foundation of agricultural production. At the same time, regions should develop mixed agriculture according to local conditions, promoting the comprehensive utilization and development of farmland and achieving a win-win situation for scale and efficiency, quality, and quantity. Third, the agricultural market should strengthen the factor input guarantee mechanism, optimize the input–output structure, improve the resource transformation efficiency, promote the efficient transformation of factors, and ensure the balance of agricultural output.

**Accelerate the process of market-oriented allocation of factors such as labor, land, energy, and water resources.** All regions should reduce the resource waste caused by the factor mismatch and imbalance of the demand and supply structure, strengthen the construction of modern agricultural facilities in the YREB, constantly improve the production capacity of agricultural development, and make the market play a decisive role in the allocation of resource factors.

**Improve the agricultural structure and mechanism and promote the transformation and upgrading of agriculture.** First, the government should strengthen the construction of agricultural parks, guide the convergence of resource elements, radiate, and drive the landing of surrounding facilities to solve problems related insufficient resources, scattered facilities, and weak industries in rural areas to improve the level of agricultural development. Second, leading enterprises should be guided to cooperate with small farmers, which will result in the implementation of the integrated development strategy for production and marketing, will constantly expand market capacity and sales channels, and will solve the problems of slow and difficult sales of agricultural products. Third, the market needs to open factor circulation channels, guide the flow of advantageous resources to weak areas, promote the free flow of factors, and realize the effective integration and sharing of regional resources.

## Figures and Tables

**Figure 1 ijerph-19-00958-f001:**
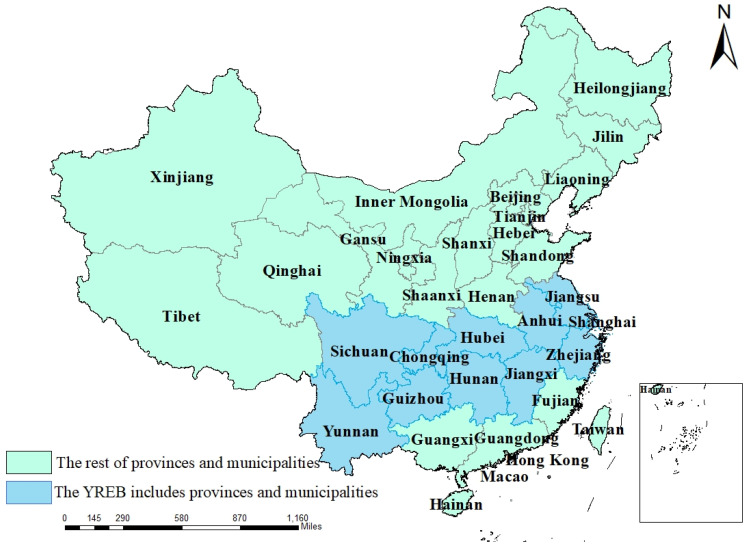
Map of China with light green shading indicating the non-YREB provinces/cities and h light blue indicating the YREB provinces/cities.

**Figure 2 ijerph-19-00958-f002:**
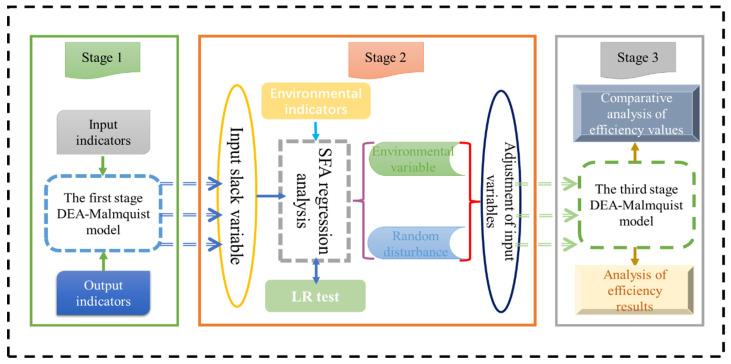
Research methods for determining the agricultural efficiency in the YREB.

**Figure 3 ijerph-19-00958-f003:**
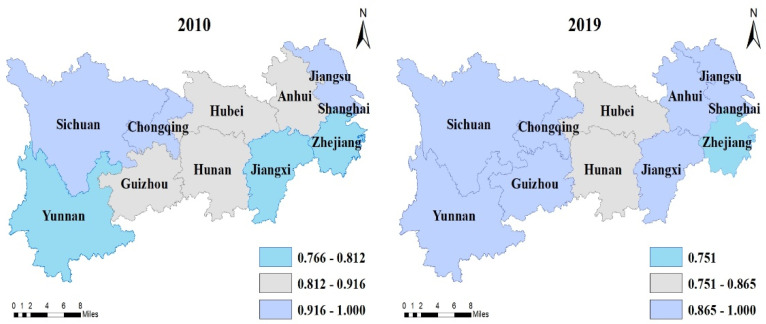
The temporal and spatial evolution of agricultural technical efficiency in the YREB.

**Figure 4 ijerph-19-00958-f004:**
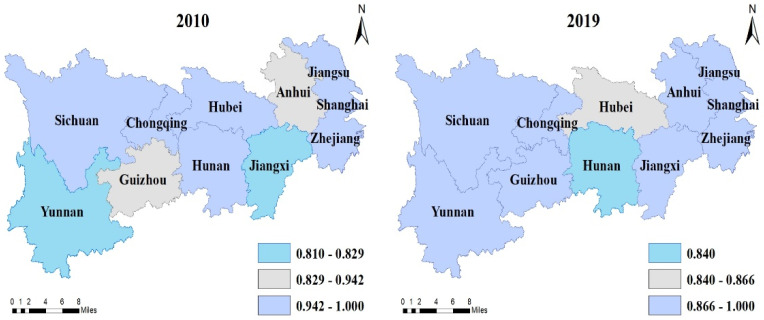
Same as Figure 3, but for pure technical efficiency in the YREB.

**Figure 5 ijerph-19-00958-f005:**
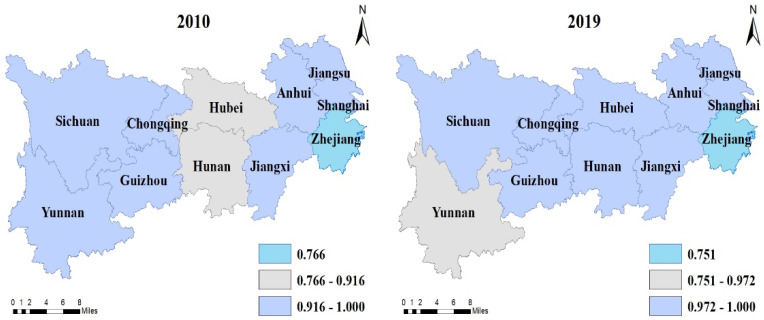
Same as Figure 3, but for scale efficiency.

**Figure 6 ijerph-19-00958-f006:**
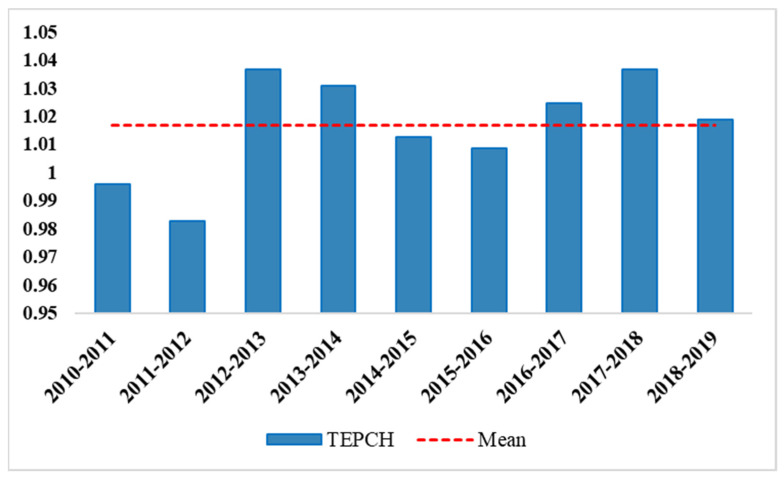
The changes in the total factor productivity (TFPCH) from 2010 to 2019.

**Figure 7 ijerph-19-00958-f007:**
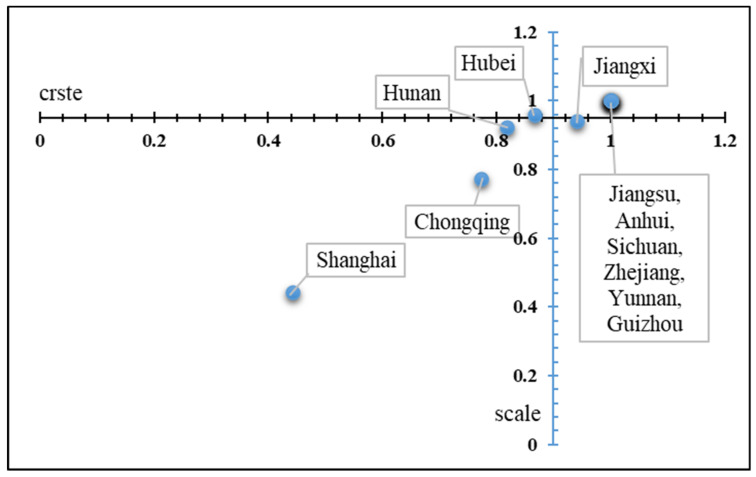
A comprehensive efficiency decomposition diagram of the YREB in 2019.

**Table 1 ijerph-19-00958-t001:** APE in 2010 and 2019.

Year	2010				2019			
Province	CRSTE	VRSTE	Scale		CRSTE	VRSTE	Scale	
Jiangsu	1	1	1	—	1	1	1	—
Anhui	0.896	0.896	1	—	1	1	1	—
Sichuan	1	1	1	—	1	1	1	—
Hubei	0.916	1	0.916	drs	0.865	0.866	0.999	drs
Chongqing	1	1	1	—	1	1	1	—
Shanghai	1	1	1	—	1	1	1	—
Zhejiang	0.766	1	0.766	drs	0.751	1	0.751	drs
Hunan	0.915	1	0.915	drs	0.838	0.84	0.999	irs
Jiangxi	0.8	0.81	0.987	irs	1	1	1	—
Yunnan	0.812	0.829	0.979	irs	0.972	1	0.972	drs
Guizhou	0.916	0.942	0.972	irs	1	1	1	—
Mean	0.911	0.953	0.958		0.948	0.973	0.975	

Note: CRSTE represents technical efficiency; VRSTE represents pure technical efficiency; scale represents scale efficiency; drs represents diminishing returns to scale; irs represents increasing returns to scale; and—represents constant returns to scale.

**Table 2 ijerph-19-00958-t002:** Evaluation of unadjusted APE in the YREB from 2010–2019.

Year	EFFCH	TECHCH	PECH	SECH	TFPCH
2010–2011	0.967	1.031	0.972	0.995	0.996
2011–2012	1.02	0.964	1.014	1.006	0.983
2012–2013	1.017	1.02	1.02	0.997	1.037
2013–2014	1.017	1.014	1.028	0.989	1.031
2014–2015	1.003	1.01	0.994	1.009	1.013
2015–2016	0.997	1.012	0.986	1.012	1.009
2016–2017	1.004	1.021	0.995	1.01	1.025
2017–2018	1.014	1.023	1.02	0.994	1.037
2018–2019	1.001	1.018	0.996	1.005	1.019
Mean	1.004	1.012	1.003	1.002	1.017

**Table 3 ijerph-19-00958-t003:** Same as Table 2, but for unadjusted APE.

Province	EFFCH	TECHCH	PECH	SECH	TFPCH
Jiangsu	1	0.99	1	1	0.99
Anhui	1.012	1.02	1.012	1	1.033
Sichuan	1	1.005	1	1	1.005
Hubei	0.994	0.999	0.984	1.01	0.993
Chongqing	1	1.008	1	1	1.008
Shanghai	1	1.007	1	1	1.007
Zhejiang	0.998	1.027	1	0.998	1.025
Hunan	0.99	1.021	0.981	1.01	1.011
Jiangxi	1.025	0.996	1.024	1.001	1.021
Yunnan	1.02	1.02	1.021	0.999	1.04
Guizhou	1.01	1.043	1.007	1.003	1.053
mean	1.004	1.012	1.003	1.002	1.017

**Table 4 ijerph-19-00958-t004:** SFA regression results.

Explanatory Variables	Explained Variable				
	A1	A2	A3	A4	A5
Constant	−196.07	−39.49	−8.31	61.79	−333.69
B1	0.027	−0.0012	−0.0005	0.046	0.04
B2	−0.049	−0.001	0.0142	0.255	0.265
B3	0.003	−0.0006	−0.0001	−0.011	0.0003
$\sigma^{2}$	135,336.16	100,262.99	402.87	401,350.78	146,890.70
γ	0.64	0.61	0.05	0.71	0.67
LR test	47.26	36.72	53.95	50.43	50.34

Note: A1 represents the total sown area of crops; A2 represents agricultural employees; A3 represents agricultural fertilizer usage; A4 represents total machinery power; A5 represents effective irrigation area; B1 represents disaster area; B2 represents agricultural subsidies; and B3 represents regional GDP.

**Table 5 ijerph-19-00958-t005:** Evaluation of adjusted APE in the YREB from 2010 to 2019.

Year	EFFCH	TECHCH	PECH	SECH	TFPCH
2010–2011	0.977	1.075	0.99	0.987	1.05
2011–2012	1.006	1.025	1.004	1.001	1.031
2012–2013	1.032	1.032	1.013	1.019	1.066
2013–2014	1.017	1.024	1.015	1.003	1.042
2014–2015	0.995	1.044	0.996	0.999	1.038
2015–2016	0.998	1.02	0.989	1.009	1.017
2016–2017	0.993	1.036	0.996	0.998	1.029
2017–2018	1.025	1.016	1.015	1.01	1.041
2018–2019	0.987	1.051	0.998	0.989	1.037
Mean	1.003	1.036	1.002	1.001	1.039

**Table 6 ijerph-19-00958-t006:** Evaluation of the adjusted APE of the provinces in the YREB from 2010 to 2019.

Province	EFFCH	TECHCH	PECH	SECH	TFPCH
Jiangsu	1	1.004	1	1	1.004
Anhui	1.009	1.021	1.008	1.001	1.031
Sichuan	1	1.033	1	1	1.033
Hubei	0.984	1.028	0.989	0.995	1.012
Chongqing	1.009	1.018	1	1.009	1.027
Shanghai	0.945	1.069	1	0.945	1.01
Zhejiang	1	1.06	1	1	1.06
Hunan	0.978	1.062	0.987	0.991	1.038
Jiangxi	1.023	0.996	1.02	1.003	1.02
Yunnan	1.034	1.052	1.014	1.02	1.088
Guizhou	1.057	1.051	1.002	1.054	1.111
Mean	1.003	1.036	1.002	1.001	1.039

**Table 7 ijerph-19-00958-t007:** Four categories of provinces/cities.

Category	Province/City
I	Shanghai, Chongqing, Hunan
II	Hubei
III	Jiangxi
IV	Jiangsu, Anhui, Sichuan, Zhejiang, Yunnan, Guizhou

## Data Availability

Publicly available datasets were analyzed in this study. This data can be found here: http://www.stats.gov.cn/ (accessed on 7 December 2021).

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
