# Peer review of "An Analysis of Agricultural Production Efficiency of Yangtze River Economic Belt Based on a Three-Stage DEA Malmquist Model"

_ijerph, 2022, doi:10.3390/ijerph19020958_

Round 1

Reviewer 1 Report

It is important to consider the efficient for food production under population growth.
The authors focus agricultural production efficiency on this manuscript. It is very interesting.
In that sense, this manuscript has some interesting results.
However, there are some of changes for the better manuscript.
Please check the comments.

p.2 figure 1 
Province border overlaps with municipality and province names in figure 1.
It is difficult to understand them. 
I suggest to make improvement.

p.6-7, figure 3 - 5 
It is difficult to understand municipality and province names on the maps and legend words in figures.
I suggest that authors show them more clearly.

Author Response

p.2 figure 1

Province border overlaps with municipality and province names in figure 1.It is difficult to understand them. I suggest to make improvement.

Response: Figure 1 has been improved.

p.6-7, figure 3 - 5

It is difficult to understand municipality and province names on the maps and legend words in figures.I suggest that authors show them more clearly.

Response: Thanks for the reviewer's suggestion, Figure 3 - 5 have been redrawn.

Reviewer 2 Report

The article is well structured and written, fluid and flows well for the reader. The methodology chosen is appropriate and the results are consistent and robust. I suggest better explaining the different variables introduced with the different equations used which for an unfamiliar reader may be difficult to understand. for example what is represented with s and what with x, y? etc.

As for the results, they are correct and robust from a technical point of view but not very explanatory. I would add some considerations and comments explaining the reasons behind it, the drivers of the efficiency differentials that have been found, what is behind the different development trajectories? etc.

Author Response

I suggest better explaining the different variables introduced with the different equations used which for an unfamiliar reader may be difficult to understand. for example what is represented with s and what with x, y? etc.

Response: Thanks to the reviewer’s suggestion, the indicator interpretation on the model has been improved.

As for the results, they are correct and robust from a technical point of view but not very explanatory. I would add some considerations and comments explaining the reasons behind it, the drivers of the efficiency differentials that have been found, what is behind the different development trajectories? etc.

Response: Thanks for the reviewer’s suggestions. The conclusions have been supplemented and improved according to the instructions.

Reviewer 3 Report

The aim of the paper is to present an analysis of agricultural production efficiency of Yangtze River Economic Belt based on a Three-stage DEA Malmquist Model.

Abstract well reflects the aim of the paper, the methodology and results.

In the Introduction, the authors should highlight the novelty of the paper. At the end of the Introduction, authors should present the main sections of the paper.

In the section 3.4, the authors should present a detailed description of the data used.

The methodology and analysis is well presented. Also, I recommend to completing section Results and discussion with the connection between the results achieved in the paper and previous results described in Literature review.

In Conclusions should be presented the potential beneficiaries of the study results.

I recommend publishing the paper after processing the suggested improvements.

Author Response

In the Introduction, the authors should highlight the novelty of the paper. At the end of the Introduction, authors should present the main sections of the paper.

Response: We have supplemented the innovation and sections of the paper in the introduction

In the section 3.4, the authors should present a detailed description of the data used.

Response: The data used includes the total sown area of crops, agricultural employees, agricultural fertilizer usage, total machinery power, effective irrigation area, disaster area, agricultural subsidies, and regional GDP. These data has been added in the section 3.4.

The methodology and analysis is well presented. Also, I recommend to completing section Results and discussion with the connection between the results achieved in the paper and previous results described in Literature review.

Response: Thanks to the reviewers for their suggestions, the results and discussion have been further analyzed in the conclusion.

In Conclusions should be presented the potential beneficiaries of the study results.

Response: The potential beneficiaries of the study results have been added in conclusion, such as the formulation of government policies and the improvement of agricultural standards.